# Treadmilling FtsZ polymers drive the directional movement of sPG-synthesis enzymes via a Brownian ratchet mechanism

Joshua W. McCausland [1], Xinxing Yang [1], Georgia R. Squyres [2], Zhixin Lyu [1], Kevin E. Bruce[3], Melissa M. Lamanna [3], Bill Söderström [4], Ethan C. Garner [2], Malcolm E. Winkler [3], Jie Xiao [1✉] & Jian Liu[5✉]

The FtsZ protein is a central component of the bacterial cell division machinery. It polymerizes at mid-cell and recruits more than 30 proteins to assemble into a macromolecular complex to direct cell wall constriction. FtsZ polymers exhibit treadmilling dynamics, driving the processive movement of enzymes that synthesize septal peptidoglycan (sPG). Here, we combine theoretical modelling with single-molecule imaging of live bacterial cells to show that FtsZ's treadmilling drives the directional movement of sPG enzymes via a Brownian ratchet mechanism. The processivity of the directional movement depends on the binding potential between FtsZ and the sPG enzyme, and on a balance between the enzyme's diffusion and FtsZ's treadmilling speed. We propose that this interplay may provide a mechanism to control the spatiotemporal distribution of active sPG enzymes, explaining the distinct roles of FtsZ treadmilling in modulating cell wall constriction rate observed in different bacteria.

[1] Department of Biophysics and Biophysical Chemistry, Johns Hopkins School of Medicine, Baltimore, MD 21205, USA. [2] Department of Molecular and Cellular Biology, Harvard University, Cambridge, MA 02138, USA. [3] Department of Biology, Indiana University Bloomington, Bloomington, IN 47405, USA. [4] The ithree Institute, University of Technology Sydney, Ultimo, NSW 2007, Australia. [5] Department of Cell Biology, Johns Hopkins School of Medicine, Baltimore, MD 21205, USA. ✉email: xiao@jhmi.edu; jliu187@jhmi.edu

During cell wall constriction in most Gram-negative bacteria, new septal peptidoglycan (sPG) synthesis and old cell wall degradation occur simultaneously[1]. A large number of the cell wall enzymes involved in this process and their regulators have been identified. However, it remains unclear how these proteins are orchestrated in time and space to achieve successful cytokinesis, and at the same time maintain the structural integrity of the septal cell wall[2,3]. Perturbations of PG remodeling at septum compromise cell division and often lead to cell lysis[4].

Recent studies have indicated that FtsZ, an essential component of the bacterial cell division machinery, may play a central part in regulating the spatiotemporal coordination of sPG synthesis enzymes. FtsZ is a highly conserved bacterial tubulin homolog and GTPase[5–7]. During cell division, FtsZ polymerizes at the cytoplasmic face of the inner membrane to form a ring-like structure (Z-ring) at mid-cell[8–10]. The Z-ring then locally recruits an ensemble of more than 30 proteins, many of which are sPG-remodeling enzymes[1,11], to initiate septal cell wall constriction. New studies employing super-resolution and single-molecule imaging in vitro and in vivo have demonstrated that the FtsZ polymers exhibit GTP hydrolysis-driven treadmilling dynamics, which are the continuous polymerization at one end and depolymerization at the other end, with individual FtsZ monomers remaining stationary in the middle[12–15]. Most interestingly, it was found that FtsZ's treadmilling dynamics drive processive movements of the essential sPG transpeptidase (TPase, FtsI in *E. coli* and PBP2B in *B. subtilis*)[12,13] and glycosyltransferase FtsW[16]. Consequently, it was proposed that FtsZ's treadmilling dynamics spatially and temporally distribute sPG synthesis enzymes along the septum plane to ensure smooth septum morphogenesis[13]. However, it is unknown how FtsZ's treadmilling dynamics with stationary monomers in the cytoplasm are transduced into the periplasm to drive the persistent and directional movement of cell wall synthesis enzymes. The role of FtsZ's treadmilling dynamics in modulating sPG synthesis activity also remains elusive, as it was shown that the cell wall constriction rate is dependent on FtsZ's treadmilling speed in *B. subtilis*[12] but not in *E. coli*[13], or *S. aureus*[17].

In this work, we combined agent-based theoretical modeling with single-molecule imaging-based experimental testing to address the mechanism of the FtsZ treadmilling-dependent processive movement of sPG enzymes, and its associated role in bacterial cell division. We found that a Brownian ratchet mechanism underlies the persistent and directional movement of single sPG synthesis enzyme molecules driven by FtsZ's treadmilling dynamics. Using FtsI as a model sPG enzyme, we found that the processivity of the Brownian ratchet is dependent on the (indirect) binding potential between FtsI and FtsZ and modulated by the balance between FtsI's random diffusion and FtsZ's treadmilling speed. This finding offers predictions about how different bacterial species could harness the same FtsZ treadmilling machinery to achieve distinct processivities of sPG enzymes, so that the available level of sPG synthases for cell wall constriction can be controlled differentially. Given the lack of linear stepping motors in the prokaryotic world, our work suggests a general framework for how polymer dynamics coupled with Brownian ratcheting could underlie directional transport of cargos, and be shaped by evolution to meet the needs of different cellular milieus.

## Results

**Model description**. Our model is based on the concept of a Brownian ratchet, where FtsZ's treadmilling introduces an asymmetry to bias the random diffusion of FtsI molecules in the periplasm, upon which FtsI persistently follows the shrinking end of a treadmilling FtsZ filament (Fig. 1). The quantitative details of the model are rooted in the physical and chemical properties of key components of the system, which can be characterized by experiments.

As shown in Fig. 1, the model describes the movement of a free FtsI molecule at the septum as quasi-1D. The model assumes that FtsI, an essential TPase with a single transmembrane domain and a cytoplasmic tail, can freely diffuse along the inner membrane at the septum or interact indirectly with a treadmilling FtsZ filament underneath the inner membrane (Fig. 1a). The dynamics of a single FtsI molecule at the septum are therefore determined by three parameters: the constant of FtsI's free diffusion ($D$), the treadmilling speed ($V_Z$) of FtsZ filaments, and the attraction force determined by the binding potential ($U$) between FtsI and FtsZ (Fig. 1b, c).

To set the ranges of the three parameters, we consider the following. First, we set the diffusion constant to range from $10^{-3}$ to $10^{-1}$ $\mu m^2/s$, which is of a typical inner membrane protein in bacterial cells[18,19]. For example, PBP2, the counterpart of FtsI in cell wall elongation, was measured at ~0.06 $\mu m^2/s$[18,19]. Second, the average treadmilling speed of FtsZ was at ~20–40 nm/s in vivo but can be a few-fold faster in vitro, therefore we set a large range of 10–100 nm/s[14,15]. Third, FtsI interacts with FtsZ at the septum

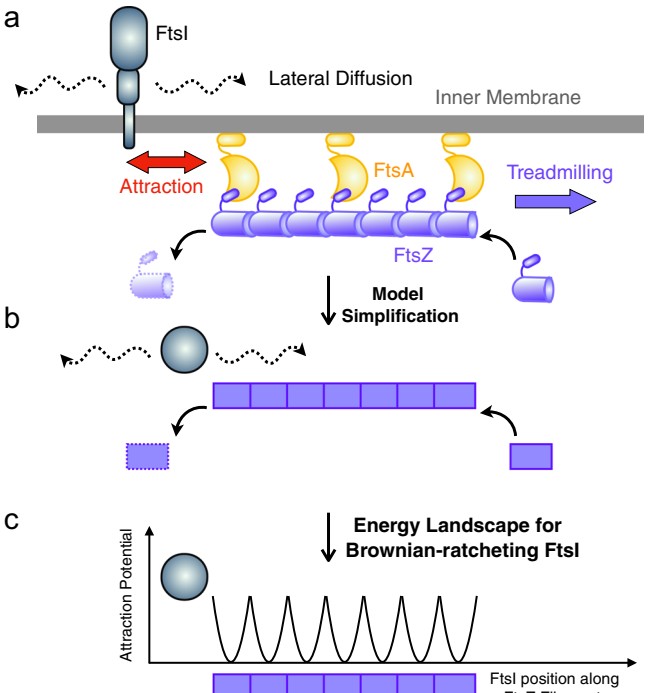

**Fig. 1 Model description. a** Schematic representation of an sPG synthase complex's interaction with FtsZ treadmilling. FtsZ resides in the cytoplasm. FtsI is a transmembrane protein that does not dissociate from the membrane even when it dissociates from FstZ. **b** Model simplification of FtsZ–FtsI interaction at the septum. The FtsZ filament (purple) undergoes treadmilling by dissociating an FtsZ subunit from the left end and associating new ones on the right end. While FtsI (gray) intrinsically diffuses around, it has a binding affinity to FtsZ subunits. **c** Schematics of FtsZ–FtsI binding potentials. Here, the binding potential is assumed to be harmonic and short-ranged (~5 nm), which is about the size of an individual FtsZ subunit. Note that there is no energy barrier for FtsI to bind to FtsZ, because the binding potential is attractive. Once FtsI binds to an FtsZ subunit, however, the binding potential presents an energetic barrier preventing FtsI from diffusing away.

indirectly through a relay of protein–protein interactions that include FtsN, FtsA, and/or FtsEX[1,20]. For simplicity we omit the details of the protein-protein interaction relay and refer to it as the interaction between FtsI and FtsZ. The indirect interaction between an FtsZ monomer and a nearby FtsI molecule constitutes an attractive force for each other, and can be described as a short-ranged harmonic binding potential (Fig. 1c). We assumed the potential range was ±2.5 nm, commensurate with the size of an FtsZ monomer (~5 nm)[21–24]. The potential's magnitude was ~10 $k_B$T, corresponding to a $K_d$ in the μM range, which is typical for protein–protein interactions in the bacterial divisome system[25–31]. Note that here we use FtsI in *E. coli* as the model sPG enzyme, but the same analysis can be applied to any other sPG enzyme or divisome proteins as well.

We first simulate the treadmilling of an FtsZ filament (Fig. 1b). The model depicts the filament shrinking and growing according to Eqs. (1) and (2), which respectively describe how the positions of the shrinking and growing ends of a treadmilling FtsZ filament at time $t$, $x_S(t)$ and $x_G(t)$ are related to the treadmilling rate, $V_Z$:

$$\frac{dx_S(t)}{dt} = V_Z, \tag{1}$$

$$\frac{dx_G(t)}{dt} = V_Z. \tag{2}$$

The model simulates the treadmilling events in a discretized sense, which occur every 5 nm/($V_Z \cdot \Delta t$) simulation time step. Each time an FtsZ subunit falls off the shrinking end of the filament, the associated binding potential vanishes with it; likewise, when an FtsZ subunit adds onto the growing end, the associated binding potential appears with it. The treadmilling speed $V_z$ of each FtsZ filament is drawn stochastically from an experimentally measured distribution[13]. Note that we do not model explicitly the stochastic on-reactions and off-reactions of individual subunits of an FtsZ filament, because such stochasticity is reflected in the treadmilling speed distribution of the filaments. The FtsZ filament length is set at 50 monomers (250 nm) and the treadmilling speed is independent of the filament length, in accordance to previous biochemical studies and a recent in vivo study[20,32–35]. To discern principal interactions, the model considers one FtsI molecule and one FtsZ filament in a self-contained septal section. It can be easily expanded to include multiple FtsI molecules per FtsZ filament (Supplementary Fig. 1).

Next, to numerically compute the model, we describe the dynamics of FtsI by a Langevin-type equation (Eq. 3), where the viscous drag force on the molecule is in balance with a driving force $f$ and a force from thermal noise $\xi$:

$$\lambda \frac{dx(t)}{dt} = f(x(t)) + \xi(t). \tag{3}$$

Here, $x(t)$ represents the location of an FtsI molecule at time $t$ along the 1D septum. $\lambda$ is the effective viscous drag coefficient for FtsI's movement with $\lambda = k_B T/D$, where $D$ is the diffusion constant of free FtsI molecules on the inner membrane when it is not interacting with FtsZ. $f(x(t))$ is the attractive force exerted upon the FtsI molecule by the FtsZ's binding potential, $U(x, t)$ (Fig. 1c). Specifically, $f(x(t)) = -\partial U(x, t)/\partial x$ at time $t$. The last term $\xi(t)$ reflects the random diffusive motion of FtsI on the inner membrane with $\langle \xi(t) \cdot \xi(t') \rangle = 2D \cdot \Delta t \cdot \delta(t-t')$, where $\Delta t$ is the unitary time step in simulation. In the simulation, the FtsI molecule diffuses after the underlying FtsZ subunit falls off from the shrinking end, as it experiences a flat 5-nm local potential. However, the FtsI molecule has a probability to diffuse to the next FtsZ subunit in the row (i.e., another ~5 nm to the right), and associate there due to the presence of the binding potential of that FtsZ subunit. It could also diffuse away and dissociate from the

FtsZ filament. This process is not deterministic but stochastic as we show below.

Finally, assuming that the FtsZ filament treadmills from left to right with a steady-state length of ~250 nm, the model implements open boundary conditions on the FtsI molecule at both the left and right edges of the system and the right-ward FtsZ treadmilling is not limited. The model results presented below reflect the nominal case, whose essence remains robust against variations of model parameters within the physical range constrained by existing experimental data.

**A Brownian ratchet links directional movement to tread-milling**. As we described above, Brownian ratcheting hinges on the diffusion of FtsI, its interaction with FtsZ, and FtsZ's treadmilling speed. To examine how the movement of FtsI depends on FtsI's diffusion and the binding potential between FtsI and FtsZ, we kept FtsZ's treadmilling speed constant at an experimentally measured speed of 25 nm/s and carried out a phase diagram study using stochastic simulations. As described above, we chose a parameter range of 0.0001 to 0.1 μm²/s for FtsI's diffusion[18,19]. The upper limit of the binding potential was set to be ~20 $k_B$T, which corresponded to a dissociation constant $K_d$ in the nM-range.

We considered an initial condition in which both the shrinking end of an FtsZ filament and an FtsI molecule were at the left boundary of the septal section. To be commensurate with our experimental analysis, we counted an FtsI trajectory as moving directionally if it tracked the shrinking end of a treadmilling FtsZ filament persistently and unidirectionally for at least 4 s. Because of the stochastic nature of Brownian ratcheting, we characterized the state of FtsI under this parameter set condition as persistent end-tracking in the phase diagram if 50% or more of simulated FtsI trajectories displayed such a persistent directional movement.

As shown in the phase diagram in Fig. 2a, the model revealed that when the binding potential between FtsZ and FtsI was weak (<5 $k_B$T, ~mM Kd), FtsI largely displayed random diffusion without directional movements along the septum. When the attraction potential was sufficiently strong (>5 $k_B$T), strong binding quenched free diffusion and confined FtsI to the end of an FtsZ filament. As the FtsZ subunit at the shrinking end of the filament fell off, FtsI was free to diffuse. However, the next FtsZ in the row presented the binding potential just a short distance away (~5 nm). By chance, the FtsI molecule diffused closer to the new end subunit of the FtsZ filament, where it was trapped again (the inset of Fig. 2b). Effectively, FtsI was "pulled" to the right by ~5 nm. With the subsequent FtsZ subunits falling off one after the other from the shrinking end, the FtsI molecule ratcheted forward and persistently tracked the end of the treadmilling FtsZ filament. These consecutive movements resulted in a persistent and directional trajectory of FtsI (Fig. 2b). Note that the directional movement of FtsI is not deterministic but rather probabilistic due to the stochastic nature of Brownian ratcheting; an FtsI molecule may decouple from a treadmilling FtsZ filament at any time (e.g., at a time point of ~14 s in the sample trajectory shown in Fig. 2b). Nevertheless, the average speed of FtsI molecules determined from their directional movement was tightly coupled to FtsZ's treadmilling speed (Fig. 2c), recapitulating the experimentally measured near-linear correlation between FtsI's directional motion with FtsZ's treadmilling speeds in both wildtype and FtsZ GTPase mutants[13].

The phase diagram (Fig. 2a) also showed that at a constant binding potential between FtsZ and FtsI, persistent end-tracking of FtsI required an appropriate range of diffusion constants. If FtsI diffused too rapidly, it could not be confined by the binding potential of the shrinking end of the FtsZ filament. Conversely,

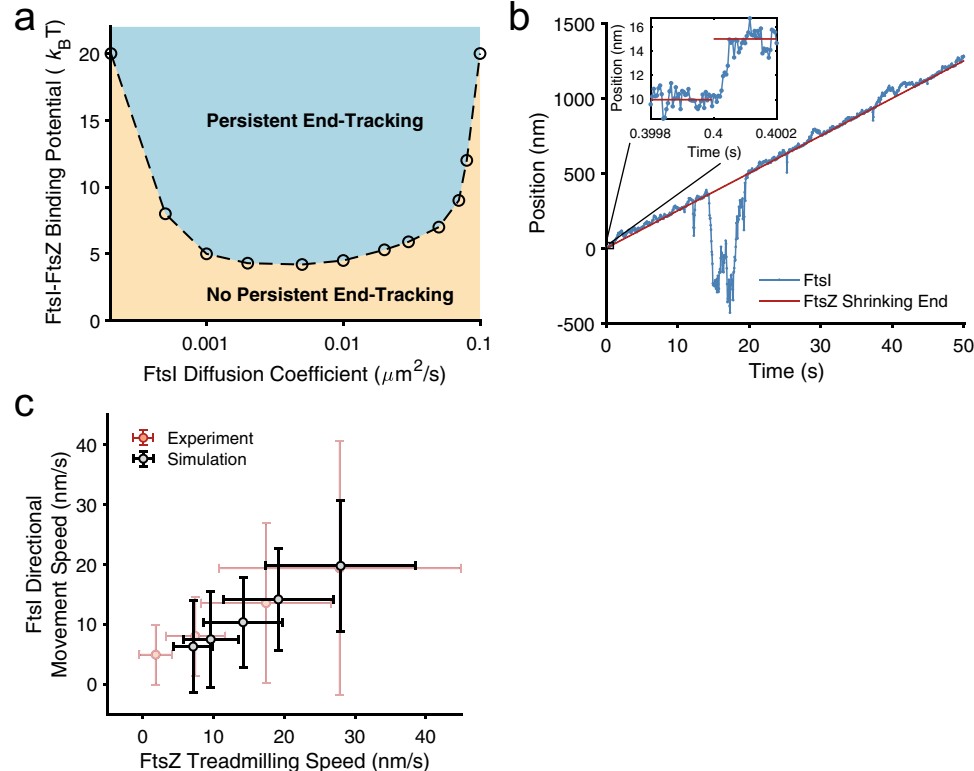

**Fig. 2 An FtsZ treadmilling-mediated Brownian ratchet mechanism drives FtsI's directional movement. a** Phase diagram depicting the dependence of FtsI's persistent end-tracking (blue-shaded region) on FtsI's diffusion constant and FtsI–FtsZ binding potential. **b** A representative simulated trajectory of an FtsI molecule persistently end-tracking with a treadmilling FtsZ filament. Inset: zoomed-in view of a boxed region of the trajectory. The model parameters for this simulation are: FtsZ's treadmilling speed $V_Z = 25$ nm/s, FtsI diffusion constant $D = 0.04$ μm²/s, FtsZ–FtsI binding potential $U = 10$ $k_B$T, and the simulation time step $= 5 \times 10^{-6}$ s. The full trajectory is plotted every $5 \times 10^{-2}$ s, and the zoomed-in inset is plotted every $5 \times 10^{-6}$ s. **c** FtsI's directional speed tightly couples with FtsZ's treadmilling speed. Each of the data points was the average of >80 independent stochastic simulation trajectories using the segments that undergo directional movement. FtsZ's treadmilling speed was fixed within each trajectory but varied across the ensemble following a Gaussian distribution with a standard deviation (SD) of 30%, in line with the experimental measurements in FtsZ WT and GTPase mutants[13,16].

when FtsI diffused too slowly, it was not able to keep up with the speed of departing FtsZ subunits at the shrinking end. Once it fell behind, the FtsI molecule lost contact with the left most FtsZ subunits permanently.

We note that an alternative initial condition, in which an FtsI molecule binds in the middle of an FtsZ filament, will result in the same end-tracking behavior. The bound FtsI molecule, if it has not dissociated, will start end-tracking when the shrinking end of the FtsZ polymer approaches and mobilizes it (Supplementary Fig. 1).

Our modeling also found that coupling to the growing end of an FtsZ filament is unable to produce the directional movement of FtsI molecules (Supplementary Fig. 2). As there is no biochemical evidence showing that the FtsI-binding potential of a newly added FtsZ subunit at the growing tip will be higher than the ones in the middle of the filament, the addition of a new FtsZ subunit at the growing end does not bias the diffusion of the FtsI molecule bound at the original tip to dissociate and re-associate with the new FtsZ subunit. Consequently, slow-diffusing FtsI molecules will be stuck in the local binding potential, unable to catch up with the addition of new FtsZ subunits (Supplementary Fig. 2a), whereas fast-diffusing FtsI molecules have a high probability of escaping from the tip, because there are no FtsZ subunits beyond the growing tip to keep it within the vicinity as that in the shrinking tip-tracking scenario (Supplementary Fig. 2b). Therefore, FtsI cannot persistently track the growing end of an FtsZ filament.

Taken together, our analysis showed that the end-tracking Brownian ratchet mechanism was able to couple FtsI's directional movement to FtsZ's shrinking end within the parameter range that is well consistent with experimentally measured data. Furthermore, the same model could explain the nondirectional movement of the cytoplasmic tail of FtsN, another divisome protein, in a recent in vitro study[25]. In this study, the cytoplasmic tail of FtsN was reported to follow the tracks of treadmilling FtsZ filaments on a supported lipid bilayer at the ensemble level. At the single molecule level, however, the FtsN tail only binds and unbinds FtsZ filaments transiently but does not exhibit directional movement[25]. Such a scenario could be explained by to our Brownian ratchet model in that the diffusion of a free FtsN cytoplasmic tail anchored on the membrane was too large (0.3–0.6 μm²/s)[25].

**FtsZ treadmilling speed modulates FtsI's end-tracking processivity.** Next, we investigated how FtsZ's treadmilling speed impacts the processivity of FtsI's directional movement at the shrinking end. Addressing this question will help us understand the role of FtsZ's treadmilling dynamics in the spatial organization and/or regulation of sPG synthesis activity. We focused on three features that collectively define the processivity of FtsI's end-tracking: (1) the propensity, (2) the run distance, and (3) the duration time of persistent end-tracking trajectories.

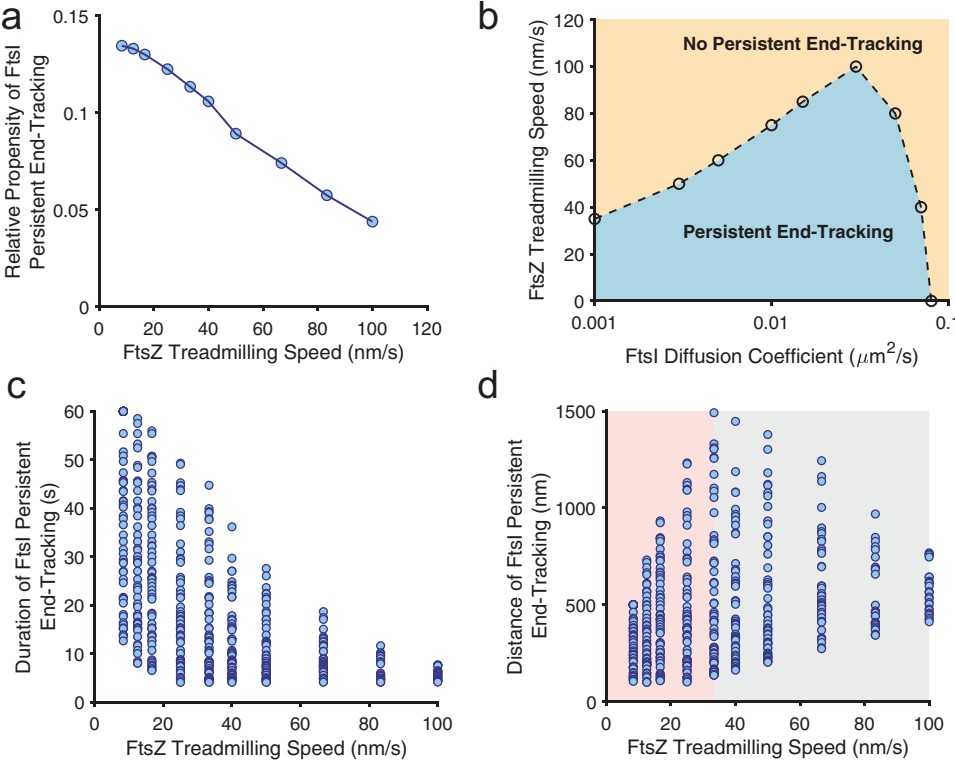

**Fig. 3 Model predictions for the processivity of FtsI directional movement modulated by FtsZ treadmilling speed. a** Predicted relative propensity of FtsI persistent end-tracking as a function of FtsZ's treadmilling speed. For each FtsZ treadmilling speed, 100 independent stochastic FtsI trajectories were simulated, from which the number of FtsI persistent end-tracking trajectories were counted. In line with our experimental measurements, the criteria for persistent end-tracking were as follows: (1) the distance between the FtsZ shrinking end and FtsI is less than 100 nm and (2) FtsI persistently follows the FtsZ shrinking end for greater than 4 s. A total of 664 out of 1000 simulated FtsI trajectories were scored as persistent end-tracking, and the number of trajectories at each FtsZ treadmilling speed was normalized as a relative probability of end-tracking. **b** Calculated phase diagram of FtsI persistent end-tracking characterized by FtsI diffusion constant and FtsZ treadmilling speed. **c** Predicted FtsZ treadmilling speed-dependence of the duration of FtsI persistent end-tracking. **d** Predicted FtsZ treadmilling speed-dependence of run distance of FtsI persistent end-tracking. The shaded regions emphasize the rise and decay of the maximum persistent run distance dependent on FtsZ's treadmilling speed. For the model calculations in **a–d**, the FtsZ–FtsI binding potential is set to be 10 $k_B$T.

We first examined how the relative propensity of FtsI's persistent end-tracking was modulated by FtsZ treadmilling speed. The relative propensity is defined as the percentage of the number of FtsI persistent end-tracking trajectories at each FtsZ treadmilling speed, normalized by the total number of FtsI persistent end-tracking trajectories of all the simulated FtsZ treadmilling speeds. Keeping the diffusion constant of FtsI at 0.04 $\mu$m$^2$/s and the binding potential at 10 $k_B$T, stochastic simulations of the Brownian ratchet model predicted that the relative propensity of persistent end-tracking trajectories of FtsI dropped off with increasing FtsZ's treadmilling speed (Fig. 3a). That is, when FtsZ treadmills too fast, FtsI could not persistently track the FtsZ shrinking end in most cases and became largely diffusive.

To further this point, we calculated the phase diagram of FtsI's persistent end-tracking propensity as a function of both FtsI's diffusion constant and FtsZ's treadmilling speed (Fig. 3b), while keeping the binding potential fixed at 10 $k_B$T. Again, we used a threshold of 50% FtsI persistent end-tracking trajectories as the criterion for the phase boundary. As shown in Fig. 3b, for a fixed diffusion constant of FtsI, there was an upper limit of FtsZ's treadmilling speed that FtsI could persistently track. Conversely, for a fixed FtsZ treadmilling speed, persistent end-tracking of FtsI required an appropriate range of diffusion constants. Importantly, very large diffusion constants of FtsI (>0.1 $\mu$m$^2$/s) did not support persistent end-tracking irrespective of FtsZ's treadmilling speed. These results were consistent with the phase diagram in

Fig. 2a and again the recent in vitro study of FtsN's cytoplasmic tail[25].

Next, we investigated how FtsZ's treadmilling speed modulates the run distance and duration time of FtsI's persistent end-tracking. The Brownian ratchet model predicted that both the run length and duration time of FtsI's persistent end-tracking should display broad distributions, due to the stochastic nature of FtsI's diffusion and the interaction between FtsI and FtsZ. Moreover, the model predicts that when FtsZ's treadmilling speed increases, the duration time of FtsI's persistent end-tracking will decrease (Fig. 3c), whereas the run distance will display a biphasic dependence—it increases to peak around an intermediate FtsZ treadmilling speed (~30 nm/s at the current parameter setting), and then decreases when FtsZ's treadmilling speed increases further (Fig. 3d). Importantly, such distinctive dependences of duration time and run distance on FtsZ's treadmilling speed is a natural consequence of the Brownian ratchet mechanism (see "Methods" section).

Qualitatively speaking, when an FtsZ subunit falls off to the cytoplasm from the shrinking end of the FtsZ filament, the associated FtsI molecule will dissociate from the FtsZ subunit, either diffuse away on the membrane, or catch up with the next FtsZ subunit in the row to continue end-tracking, the latter depending on how fast FtsZ treadmills. When FtsZ treadmills too fast (for example >30 nm/s), it will be difficult for FtsI to catch up (Fig. 3a), resulting in early termination of end-tracking, and

hence both the persistent run distance and duration time will be short (right sides of Fig. 3c, d). When FtsZ treadmills relatively slowly (<30 nm/s), the probability of FtsI catching up with the shrinking end of the FtsZ filament is high (Fig. 3a). Therefore, the slower FtsZ treadmills, the fewer number of dissociation events an end-tracking FtsI molecule would face, and hence the lower the chance for FtsI to diffuse away, leading to a longer time duration of the persistent run (Fig. 3c). Within the same time window, however, the persistent run distance will be proportional to FtsZ's treadmilling speed as predicted in Fig. 3d, that is, the slower FtsZ treadmills, the shorter FtsI's persistent run distance is. One can imagine in one extreme case where FtsZ does not treadmill at all ($V_z = 0$), the duration time of persistent runs would then be the longest and mainly dictated by the intrinsic dissociation rate of FtsI from FtsZ, and the persistent run distance would be the shortest (i.e., the size of a single FtsZ subunit). In the other extreme case where FtsZ treadmills infinitely fast ($V_z = \infty$), the duration time and distance of persistent runs would both be zero because FtsI could never track it. An analytical proof of these relationships is provided in the "Methods" section and Supplementary Fig. 3.

## Single-molecule tracking of FtsI confirms model predictions.

To experimentally examine the model's predictions on the modulation of the processivity of FtsI's directional movement by FtsZ's treadmilling speed, we performed single-molecule tracking (SMT) of a functional sandwich fusion protein Halo-FtsI[SW] labeled with JF646 in live *E. coli* cells[36,37]. To avoid disrupting the cytoplasmic interactions of FtsI's N-terminal tail with other divisome proteins, we inserted the Halo tag between the last residue (18) of the N-terminal cytoplasmic tail and the first residue of the inner membrane helix (19) of FtsI (Fig. 4a). We integrated the *halo-ftsI*[SW] fusion gene into the chromosome replacing the endogenous *ftsI* gene, and showed that it was expressed as a full-length fusion protein and supported normal cell division as a sole cellular source of FtsI similar to wild-type (WT) cells (Supplementary Fig. 4 and Supplementary Table 2).

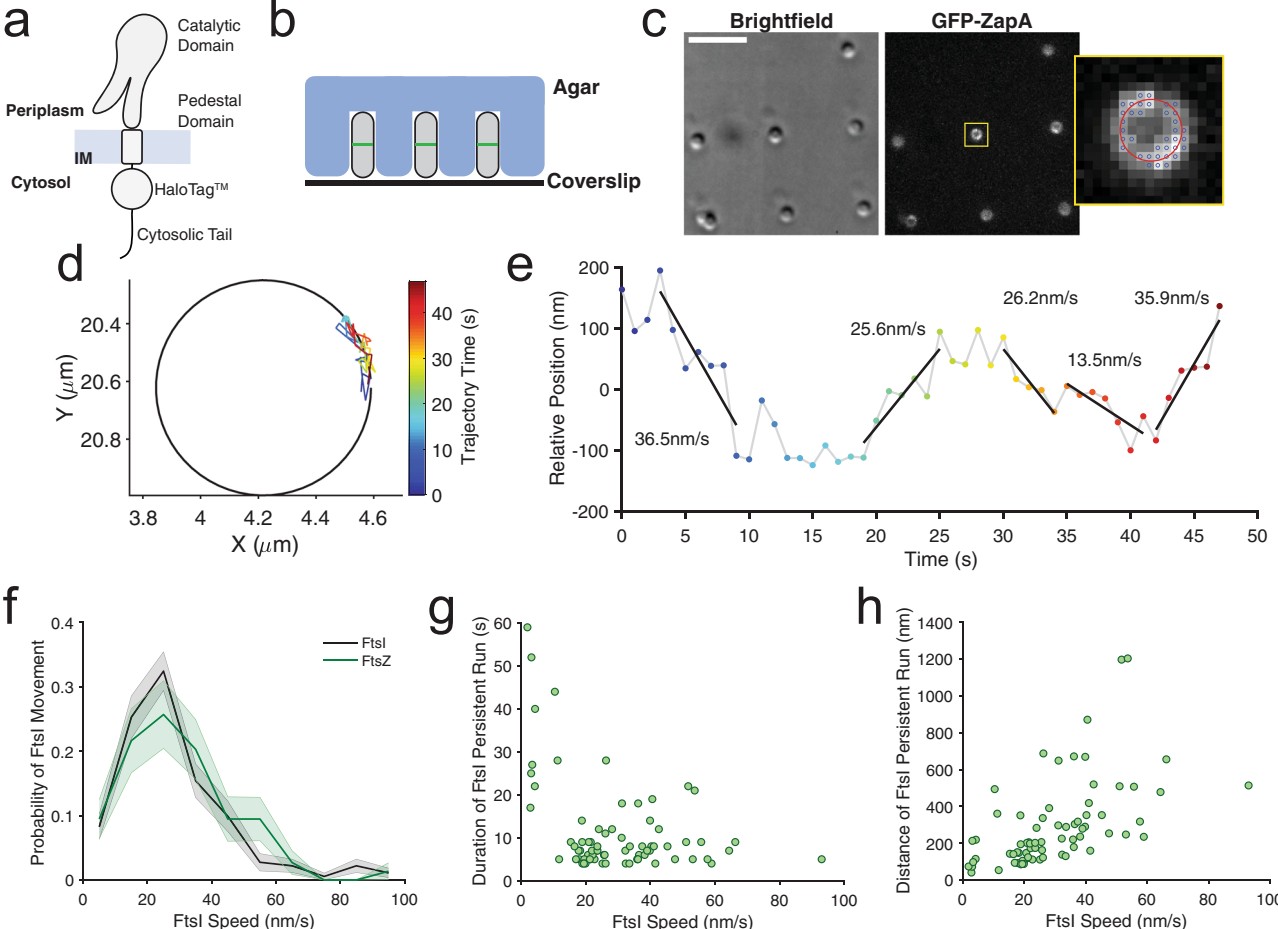

**Fig. 4 Experimental characterization of FtsI directional motion. a** Schematic of the functional sandwich fusion of FtsI. The Halo tag is inserted between residues 18 and 19 of FtsI, immediately before the first residue of the transmembrane (TM) domain. **b** Diagram of individual *E. coli* cells loaded in microholes made by nanopillars. **c** Brightfield and fluorescence images of microholes loaded with *E. coli* cells labeled with GFP-ZapA and Halo-FtsI[SW] fusion proteins. Inset shows the zoomed image of one cell in the yellow box. Blue circles indicated the pixels used for the fit of ZapA-GFP circle. Scale bar is 5 μm. Experiment repeated 18 times with similar results. **d** Circle-fitting of GFP-ZapA image super-imposed with the trajectory of a single FtsI molecule, colored in time. **e** the unwrapped trajectory from D with fitted lines at each segment to extract directional speeds, persistent run duration and distance. **f** Distributions of FtsI's directional movement speeds (*n* = 77 trajectories, 74 directional events) and FtsZ's treadmilling speed (from Yang et al.[13]). For the purpose of this study, we only plotted the fast-moving population of FtsI that follows FtsZ's treadmilling and leave out the slow-moving population of FtsI that we show to be independent of FtsZ's treadmilling[13]. Both FtsI's and FtsZ's histograms were bootstrapped 100 times to provide the shaded standard error bars (mean ± S.E.M.). A two-sample Kolmogrov–Smirnov test comparing the two datasets showed no difference in their distributions (*p* = 0.1852). **g** Dependence of the duration of FtsI's persistent run on its speed. **h** Dependence of the distance of FtsI's persistent run on its speed.

To obtain precise measurements of the persistent run distance and duration time of single Halo-FtsI[SW] molecules, we trapped individual *E. coli* cells vertically in agarose microholes made using cell-shaped nanopillar molds as previously described[38,39], so that the entire circumference of the septum could be visualized at the same focal plane (Fig. 4b). To determine whether a Halo-FtsI[SW] molecule was at a septum, we labeled the FtsZ-ring using an ectopically expressed GFP-ZapA fusion protein, which we and others have previously shown as a faithful marker of Z-ring localization and dynamics[40]. The GFP-ZapA image also allowed us to unwrap the circular trajectories of FtsI-Halo molecules along the septum (Fig. 4d) to linear displacements along the circumference of the septum (Fig. 4e), from which we could measure the persistent run speed, distance, and duration time (Fig. 4f, g, h).

As shown in Fig. 4f, the directional motion speed of Halo-FtsI exhibited a wide distribution, similar to what we previously observed for FtsZ's treadmilling. The similarity between FtsI's directional motion speed distribution and FtsZ's treadmilling speed distribution suggests that at these speed ranges, *E. coli* FtsI can faithfully end-track treadmilling FtsZ filaments as the model predicted, a point that will become important in the section below. Most importantly, the persistence run distance and duration time exhibited largely the same trends as what were predicted by the model: while the run duration time decreased monotonically (Fig. 4g), the persistence run distance increased and then decreased when FtsI's speed increased (Fig. 4h). Note here that we inferred FtsZ's treadmilling speed from FtsI's directional motion speed due to the difficulty of a two-color co-tracking experiment in the same cells, and because we have demonstrated previously that these two were linearly coupled[13]. Another potential caveat in these experiments was that a very fast FtsZ treadmilling speed (i.e., >80 nm/s) is rare in wildtype *E. coli* cells as we showed previously. Therefore, given the relatively small dataset for high speed FtsZ treadmilling, our data cannot definitively determine whether FtsI could effectively end-track FtsZ filaments of very fast treadmilling speeds. Nevertheless, the agreement of our experimental measurements with theoretical predictions supported the validity of the Brownian ratchet model.

**The dependence of sPG enzymatic activity on FtsZ treadmilling.** In *E. coli*, the total amount of septal PG synthesis and the septum constriction rate are insensitive to perturbations in FtsZ's treadmilling speed from ~8 to ~30 nm/s in a series of FtsZ GTPase mutants[13]. This insensitivity suggests that end-tracking, directionally-moving FtsI molecules were inactive in sPG synthesis. Indeed, a second, slow-moving population of FtsW and FtsI (~8 nm/s) is found to move independently of FtsZ's treadmilling, and likely corresponds to the active population of sPG synthesis in *E. coli*[16]. Similarly, in. *S. pneumoniae*, FtsW and its cognate TPase PBP2x were found to move completely independently of FtsZ's treadmilling[41], likely representing the active population of sPG synthase as that in *E. coli*. In *B. subtilis*, however, it was shown that the cell wall constriction speed is positively correlated with FtsZ's treadmilling speed, suggesting that the faster FtsZ treadmills, the higher the sPG synthesis activity[12]. How could the same FtsZ treadmilling dynamics result in different sPG synthesis activity in different species?

We propose that since FtsI molecules tracking FtsZ filaments are most likely inactive[16], the population of FtsI molecules not tracking with treadmilling FtsZ polymers is then available for sPG synthesis. Therefore, FtsI's off-rate, or the reciprocal of the time an FtsI molecule spends bound in the middle of FtsZ polymers and/or persistently end-tracking, represents the rate at which an FtsI molecule becomes available for sPG synthesis, and therefore

is proportional to the sPG synthesis rate. As such, FtsZ's treadmilling speed could modulate the rate of sPG synthesis in different bacterial species depending on the unique combination of the enzyme's diffusion coefficient and binding potentials. This modulation could explain the difference observed between *E. coli* and *B. subtilis*.

As shown in Fig. 5a, the model suggests that when FtsI diffuses relatively fast (~0.05 µm²/s, blue line), the lifetime of FtsZ-bound FtsI is largely insensitive to FtsZ's treadmilling speed in a 3-fold range from ~8 to 25 nm/s. In contrast, when FtsI diffuses relatively slowly, the lifetime of FtsZ-bound FtsI is critically dependent on FtsZ treadmilling speed. For example, at a diffusion constant of 0.005 µm²/s, the relative lifetime of FtsI molecules decreased by ~70% when FtsZ treadmilling speed increased from ~8 to 25 nm/s (Fig. 5a, green line).

The physical reason behind this drastic difference between fast and slow FtsI diffusion lies at the core of Brownian ratchet mechanism. Fast diffusion will allow FtsI to catch up with the shrinking end of an FtsZ filament in a very short time (Fig. 5b). When FtsI's diffusion becomes slower and slower, it eventually becomes the rate-limiting factor in the Brownian ratchet—a slow FtsI molecule falls behind the FtsZ shrinking end and takes a long time to catch up with the departing FtsZ filament, or simply diffuses away and become lost (Fig. 5c). As such, further increasing the FtsZ treadmilling speed in the latter case will significantly reduce the chance of FtsI keeping up with the FtsZ shrinking end and, hence the lifetime of the FtsZ-bound FtsI. Crucially, this diffusion-modulated dependence of sPG synthesis on FtsZ treadmilling speed hinges on the binding between FtsI and FtsZ. When the binding potential is reduced (i.e., weaker binding), the lifetime of FtsZ-bound FtsI is less sensitive to FtsZ speed than its higher-potential counterpart (compare Fig. 5a, d, e). Therefore, how FtsZ's treadmilling speed modulates the rate of sPG synthesis depends on the combined effects of the sPG synthase's diffusion coefficient and FtsZ-binding potential.

To examine this hypothesis, we performed fast frame-rate single-molecule tracking to measure the diffusion coefficients of free sPG synthase molecules outside the septum (FtsI in *E. coli* and PBP2b in *B. subtilis*, Fig. 5f). As FtsZ only localizes to mid-cell during cell division, sPG synthase molecules not localized to mid-cell are considered free and not interacting with FtsZ. Using the measured diffusion coefficients, we then fit the experimentally observed dependence of cell wall constriction rate on FtsZ treadmilling speed in *E. coli* and *B. subtilis*[12,42] with the normalized off-rate calculated from the model. The only free parameter in the model fitting is the binding potential, which was not possible to measure accurately in live cells with available experimental methods. As shown in Fig. 5g, h, we found that with the apparent diffusion constants of FtsI in *E. coli* and PBP2b in *B. subtilis* measured at ~0.041 ± 0.0051 (mean ± S.E.M., N = 5049 trajectories) and 0.038 ± 0.0019 µm²/s (mean ± S.E.M., N = 6765 trajectories) respectively, and binding potentials set at 8–9 and 10–12 $k_B$T, respectively, the model quantitatively recapitulated the differential dependence of cell wall constriction rate on FtsZ's treadmilling speed in the two species as previously measured. The higher binding potential of PBP2b to FtsZ in *B. subtilis*, likely due to the significantly different protein–protein interactions in Gram-positive bacteria, renders tighter coupling between end-tracking PBP2b molecules with FtsZ than that in *E. coli*, hence the fraction of end-tracking FtsI can be sensitively modulated by FtsZ's treadmilling speed in *B. subtilis*, but that of FtsI in *E. coli* cannot.

As a comparison, we also measured the diffusion of FtsW in *S. pneumoniae* (Fig. 5f). We found that the apparent diffusion

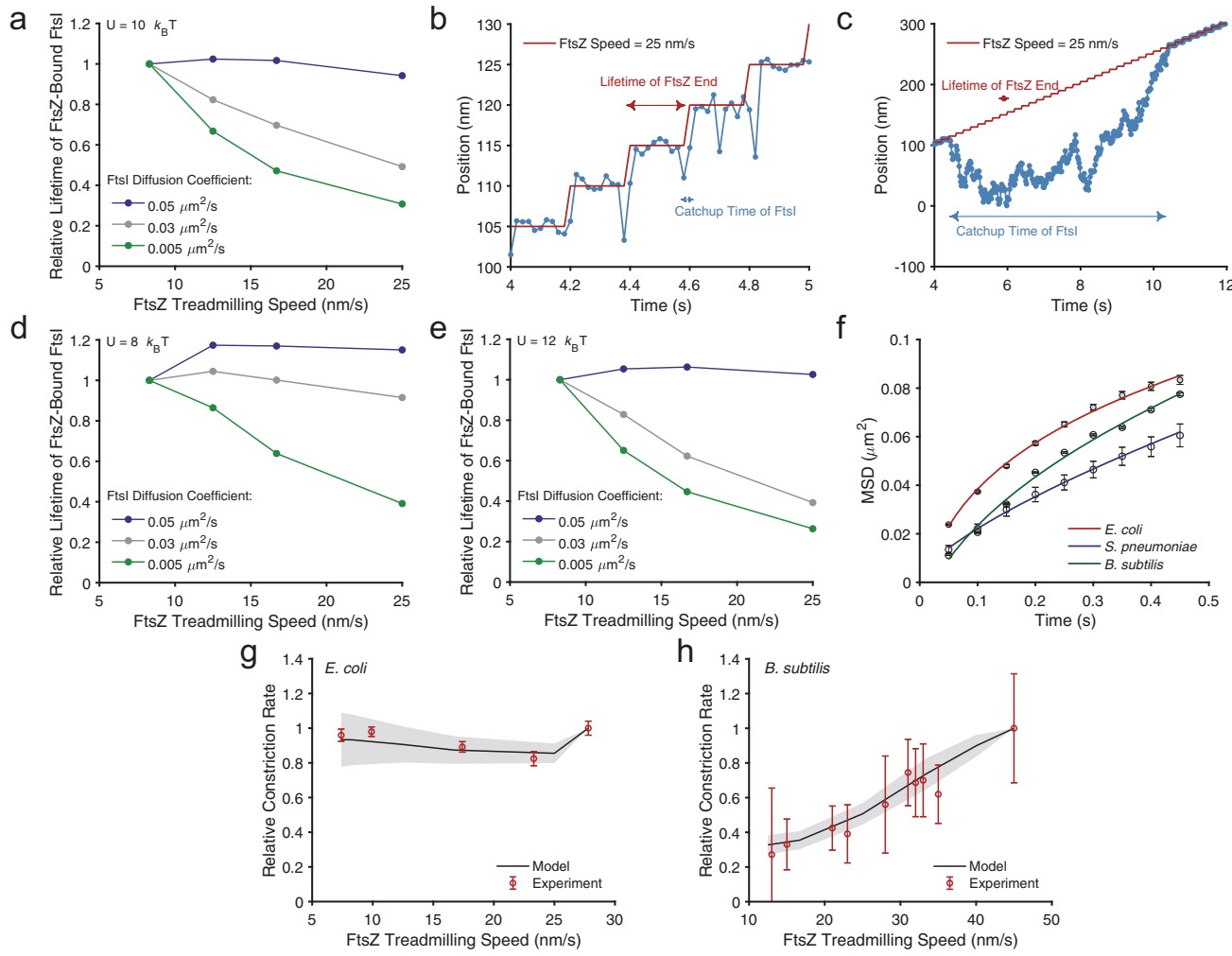

**Fig. 5 Diffusion and binding potential modulate sPG synthase's processivity on FtsZ's treadmilling speed. a** Predicted dependence of FtsZ-bound FtsI lifetime on FtsZ treadmilling speed with a binding potential of 10 $k_B$T at different FtsI diffusion coefficients. **b** A representative trajectory of persistent end-tracking when FtsI diffusion is fast (0.05 µm²/s). **c** A representative trajectory of persistent end-tracking when FtsI diffusion is slow (0.005 µm²/s). **d** Similar to **a**, but with a weaker binding potential (8 $k_B$T) at different FtsI diffusion coefficients. **e** Similar to **a**, but with a stronger binding potential (12 $k_B$T) at different FtsI diffusion coefficients. For **a**, **d**, and **e**, FtsI is initially randomly positioned along an FtsZ filament. The lifetime of FtsZ-bound FtsI was defined as the first time that FtsI escapes from either end of the FtsZ filament for greater than 100 nm. For **b** and **c**, the model results are plotted every $2 \times 10^{-2}$ s with a simulation time step of $10^{-5}$ s. **f** Mean-squared displacement (MSD) of FtsI, PBP2b and FtsW outside the septum in wildtype *E. coli*, *B. subtilis*, and *S. pneumoniae*. The fitted diffusion constants of FtsI, PBP2b, and FtsW were 0.041, 0.038, and 0.028 µm²/s, with $\alpha$ = 0.29, 0.51, and 0.71, respectively. Error bars represent standard error of the mean. **g** Model fitting of the dependence of sPG synthesis activity on FtsZ treadmilling speed in *E. coli*. The relative experimental data of sPG synthesis activity was taken from previous constriction rate measurements in Coltharp et al.[42] and Yang et al.[13] (mean ± S.E.M., each paper reported at least $n > 30$ cells). **h** Model fitting of the dependence of sPG synthesis activity on FtsZ treadmilling speed in *B. subtilis*. The relative experimental data of sPG synthesis activity was taken from previous cell division time measurements in Bisson-Filho et al.[12] (mean ± standard deviation. $n = 100$ cells for cell constriction measurements, $n > 50$ cells for treadmilling measurements). For **g** and **h**, the model fitting used the measured diffusion constants from **f**, with varying binding potentials determined by lower and upper limits (*E. coli*: 8 and 9 $k_B$T; *B. subtilis*: 10 and 12 $k_B$T), marking the boundaries of the shaded regions.

coefficient of FtsW was at $0.028 \pm 0.0004$ µm²/s (mean ± S.E.M., $N = 21$ trajectories), in the same order of magnitude as that of *E. coli* and *B. subtilis*. As FtsW does not follow the treadmilling of FtsZ at all in *S. pneumoniae*, it is most likely that the binding potential between FtsW and FtsZ is significantly lower than 5 $K_B$T as predicted by the model (Fig. 2a) under the experimental condition. It remains interesting to investigate in *S. pneumoniae* whether other divisome proteins are also independent of FtsZ's treadmilling, or they exhibit conditional dependence once the protein–protein interactions of the divisome are altered due to the presence or depletion of their binding partners. These possibilities will be further investigated in our future work.

## Discussion

In this work, we presented data supporting a Brownian ratchet model that couples the directional movements of sPG synthases to FtsZ's treadmilling, and underlies the differential sensitivity of sPG synthesis to FtsZ's treadmilling speed in *E. coli* and *B. subtilis*.

We first show that an sPG synthase molecule (here using FtsI as the model enzyme) can follow a treadmilling FtsZ polymer by end-tracking its shrinking tip, but not the growing tip, due to the intricate interplay between the sPG synthase molecule's diffusion and its binding potential to FtsZ. Only within a particular diffusion range (~0.001–0.1 µm²/s) and at a sufficient binding

potential ($>5\ k_{\mathrm{B}}T$), an sPG synthase molecule can exhibit FtsZ-treadmilling-dependent directional movement (Fig. 2). Furthermore, we show that the persistent run duration and distance of FtsI exhibit different dependence on FtsZ's treadmilling speed, as predicted by the Brownian ratchet model (Fig. 3). Using single-molecule tracking, we confirmed these model predictions (Fig. 4). The ability of treadmilling FtsZ polymers to modulate the persistent run duration and distance of sPG synthase molecules could play an important role in regulating the spatial distribution of sPG synthases to ensure the correct septum shape. Finally, we show that the Brownian ratchet model could explain the differential dependence of sPG synthesis activity on FtsZ's treadmilling speed in *E. coli*, *B. subtilis* and *S. pneumoniae*.

Given experimentally measured diffusion coefficients of sPG synthases in different bacterial species, the Brownian ratchet model predicts that the tighter binding between PBP2b and FtsZ in *B. subtilis* could cause tighter coupling between them. Hence, the fraction of time a PBP2b molecule spends on FtsZ, and consequently the fraction of time it is off FtsZ to become available for sPG synthesis, can be sensitively modulated by FtsZ's treadmilling speed. In *E. coli* or *S. pneumoniae*, the binding potential between FtsI/FtsW and FtsZ may not be as high as that in *B. subtilis*, and hence the fraction of time an FtsI/FtsW molecule remains end-tracking FtsZ exhibits much less sensitivity or no sensitivity at all to FtsZ's treadmilling speed. Different binding potentials between these different bacterial species are likely, as detailed molecular interactions among the septal ring complexes are distinct in each species. These results suggest that the same Brownian ratchet machinery may be at work, but operate in distinct regimes of the parameter space in different species. Consequently, sPG synthesis depends on FtsZ treadmilling differentially, reflecting different strategies to meet different functional needs.

We note that additional factors could also be at play. For example, the expression level of sPG synthase relative to that of FtsZ in *B. subtilis* is significantly higher than that in *E. coli* or *S. pneumoniae*[43–51] (Supplementary Table 1). This suggests that the number of sPG synthases per FtsZ filament in *B. subtilis* would be higher than that in *E. coli* or *S. pneumoniae*. Our Brownian ratchet model predicts that this condition further enhances the sensitivity of sPG synthesis activity to FtsZ's treadmilling speed, as sPG synthase molecules bound to the inner positions of an FtsZ polymer would "knock" the end-tracking one off FtsZ (or vice versa, Supplementary Fig. 1). Therefore, the faster FtsZ treadmills, the faster the FtsZ shrinking end catches up to the sPG synthase in the middle and the more sPG synthase molecules will be dissociated from FtsZ to become available for sPG synthesis. This is a stark contrast to the case of a single sPG synthase per FtsZ filament, which is mostly likely the case in *E. coli* or *S. pneumoniae* (Supplementary Fig. 1).

Moreover, the level of cell wall synthesis precursors, for example, could be another important factor. It is possible that across bacterial species, sPG synthase molecules are coupled to FtsZ's treadmilling dynamics and their lifetime on FtsZ can be sensitively modulated by FtsZ's treadmilling speed. However, if the level of a cell wall synthesis precursor is limiting, which is likely the case in *E. coli*, such a sensitivity could be further masked by the limited level of the precursor[52–55]. In *S. pneumoniae*, besides a low binding potential between an sPG synthase and FtsZ, cell wall synthesis precursor levels could also play a role in the independence of FtsZ's treadmilling. High enough levels of PG precursors could saturate all sPG synthase molecules so that no free ones are available to track with FtsZ polymers.

In summary, given the lack of linear stepper motors in prokaryotic world, Brownian ratcheting appears to be an ancient mechanism for directed cargo transportation in bacteria—another salient example is ParA-mediated DNA partitioning[56–58].

Interestingly, a similar Brownian ratchet mechanism also underlies the directional movement of mitotic chromosomes by end-tracking spindle microtubule in eukaryotes[59]. Can we distill unified fundamental principle(s) by which evolution shapes the same Brownian ratchet mechanism to meet distinct needs under different contexts? We will relegate these exciting questions to our future study.

## Methods

**Analytic solution**. We aim to obtain the analytical solution for the FtsZ treadmilling speed-dependence of run distance and duration of FtsI persistent end-tracking. An FtsI's persistent end-tracking trajectory can be decomposed into repeating steps, each of which consists of two consecutive processes. Process (1) is "stay-on": The FtsI molecule stays inside the binding potential of the FtsZ end subunit. Process (2) is "catch-up": The FtsI molecule catches up with the next FtsZ subunit in the row when the FtsZ end subunit dissociates. We next calculate the probabilities of the two processes.

For FtsI in the "stay-on" process (Supplementary Fig. 3a), it can undergo two reactions in parallel: FtsI can escape from the binding potential of the FtsZ end subunit with the rate of $1/\tau_{\mathrm{D}}$, and the FtsZ end subunit can dissociate with the rate of $1/\tau_{\mathrm{Z}}$. Here, $\tau_{\mathrm{Z}}$ is the average lifetime of an FtsZ subunit, inversely proportional to the FtsZ treadmilling speed, $\tau_{\mathrm{Z}} = \frac{5\,\mathrm{nm}}{V_{\mathrm{Z}}}$. $\tau_{\mathrm{D}}$ is the average duration of FtsI staying in the binding potential if the FtsZ subunit never falls off. The probability that FtsI is still in "stay-on" at time, $t$, is $P_1 = \exp\left(-\left(\frac{1}{\tau_{\mathrm{D}}} + \frac{1}{\tau_{\mathrm{Z}}}\right)t\right)$. Thus, by the time the FtsZ end subunit dissociates, $P_1$ scales as $P_1 \sim \exp\left(-\frac{\tau_{\mathrm{Z}}}{\tau_{\mathrm{D}}}\right)$. Likewise, the catch-up probability of the FtsI molecule $P_2$ scales as $P_2 \sim \exp\left(-\frac{\tau_{\mathrm{C}}}{\tau_{\mathrm{Z}}}\right)$. Here, $\tau_{\mathrm{C}}$ is the average catch-up time (Supplementary Fig. 3b). Taken together, the probability for FtsI end-tracking in each repeating step is:

$$P = P_1 P_2 \sim \exp\left(-\left(\frac{\tau_{\mathrm{Z}}}{\tau_{\mathrm{D}}} + \frac{\tau_{\mathrm{C}}}{\tau_{\mathrm{Z}}}\right)\right) \tag{4}$$

It follows that the probability of persistent end-tracking exact $N$-repeating steps is $P^N(1-P)$. Consequently, the average number of FtsI persistent end-tracking steps is:

$$\langle N \rangle = \frac{\sum_{N=0}^{\infty} N \cdot P^N \cdot (1-P)}{\sum_{N=0}^{\infty} P^N \cdot (1-P)} \tag{5}$$

Taking the continuum limit, $\langle N \rangle = \frac{\int_0^{\infty}\{N \cdot P^N \cdot (1-P)\}\mathrm{d}N}{\int_0^{\infty}\{P^N (1-P)\}\mathrm{d}N}$, which yields $\langle N \rangle = \frac{\tau_{\mathrm{D}}\tau_{\mathrm{Z}}}{\tau_{\mathrm{Z}}^2 + \tau_{\mathrm{C}}\tau_{\mathrm{D}}}$. The corresponding average run length and duration are:

$$\langle L \rangle = L_0 \frac{\tau_{\mathrm{D}}\tau_{\mathrm{Z}}}{\tau_{\mathrm{Z}}^2 + \tau_{\mathrm{C}}\tau_{\mathrm{D}}}, \tag{6}$$

and

$$\langle T \rangle = \frac{\tau_{\mathrm{D}}\tau_{\mathrm{Z}}}{\tau_{\mathrm{Z}}^2 + \tau_{\mathrm{C}}\tau_{\mathrm{D}}}\tau_{\mathrm{Z}} \tag{7}$$

Here, $L_0 = 5$ nm, the length of FtsZ subunit.

Specifically, persistent end-tracking entails that $\tau_{\mathrm{D}} \gg \tau_{\mathrm{Z}} \gg \tau_{\mathrm{C}}$. According to our numerical simulation, normally $\tau_{\mathrm{D}} \sim 60$ s and $\tau_{\mathrm{C}} \sim 0.001$ s. Within this parameter range, Eqs. 6–7 can quantitatively recapitulate the distinctive dependences of run length and duration on FtsZ treadmilling speed. Herein, Supplementary Fig. 3c presents a representative result of the analytic solution with $\tau_{\mathrm{D}} = 60$ s and $\tau_{\mathrm{C}} = 0.0003$ s.

Qualitatively speaking, the longer the lifetime of an FtsZ subunit, FtsI in process (1) will have a higher chance to escape the binding potential of an FtsZ end subunit, decreasing the stay-on probability. In contrast, a longer lifetime of an FtsZ subunit will allow a longer time for FtsI in process (2) to catch up to the next FtsZ subunit in the row, increasing the catch-up probability. Therefore, the balance between the stay-on and catch-up processes defines an optimal lifetime of an FtsZ subunit—and, hence, the optimal FtsZ treadmilling speed—that maximizes the probability of FtsI end-tracking per repeating step. This determines the maximum of the average number of repeating steps, and explains the biphasic dependence of FtsI persistent run length on FtsZ treadmilling speed. On the other hand, the duration per repeating step is approximately the lifetime of an FtsZ subunit. As the FtsZ treadmilling speed increases, the decrease in the lifetime of the FtsZ subunit outweighs the corresponding variation in the average number of repeating steps, leading to the decrease in the overall duration of persistent end-tracking.

**Media, bacterial strains, and plasmids**. Cells were grown in M9 minimal medium or lysogeny broth (LB) (10% tryptone, 10% NaCl, and 5% yeast extract). Fresh LB plates of strains were struck with appropriate antibiotics (detailed below) once-per-week from frozen stocks, and all cultures were started with a single colony. Bacterial strains and plasmids used are detailed in Supplementary Table 1.

**Growth curve**. Three biological replicates from TB28 and JM136 were grown from single colonies in M9+Glucose (M9) [1× M9 salts (Sigma-Aldrich M9956), 1× Amino Acids (Sigma-Aldrich M5550), 0.4% glucose, 1× Vitamins (Sigma-Aldrich M6895), 2 mM MgSO$_4$, 100 μM CaCl$_2$] as overnights at 30 °C. The following day, the overnights' OD$_{600}$ were measured with a nanodrop and diluted to an OD$_{600}$ of 0.1 in 200 μl M9 using a Corning Costar sterile 96-well plate. The 96-well plate was incubated in a Tecan Infinite M200 Pro set at 30 °C, where it would measure the OD$_{600}$ of designated wells once every 30 min for 23.5 h, shaking the plate for 3 min at 220 rpm before measuring. To obtain the growth rate, the linear phase of the log-transformed growth curve data was fitted to a straight line. The slope of that line was used to calculate the doubling time through the equation below.

$$\text{Doubling time} = \frac{\log 10(2)}{\text{slope}}$$

**Western blot of FtsI**. TB28 and JM136 were grown as overnights from single colonies in M9 at 37 °C. They were propagated 1:50 in 3 ml M9 and grown an additional 16 h at 25 °C until their OD$_{600}$ reached ~0.5, at which point 500 μl was harvested. These samples were pelleted, flash-frozen, and stored at −80 °C for one hour. These pellets were then resuspended with PBS and mixed with 2× SDS buffer (100 mM Tris-HCL, 4% SDS, 0.2% bromophenol blue, 20% glycerol, 100 mM DTT). Cells were incubated for 10 min at 95 °C in a thermocycler, then 20 μl of each were loaded into a 10% BioRad polyacrylamide gel with 5 μl of PageRuler as a ladder. After electrophoresis, the gel was transferred at 25 V for 2 h to a nitrocellulose membrane. Membranes were checked for even transfer with a Ponceau stain before blotting. Rabbit α-FtsI, generously donated by Dr. David Weiss, was used as the primary antibody with a 1/50,000 dilution. Blots were then stained with 1/50,000 HRP goat antirabbit secondary antibody. Bands were visualized with a Clarity Western ECL Substrate BioRad kit and recorded using Blu-C autoradiography film for 10 s. This western was repeated twice with four more biological replicates overall with similar results.

**Guide RNA and repair oligo design for the Halo-FtsI$^{SW}$ fusion**. The chromosomal *ftsI* gene coordinates were first located using EcoCyc, which were then used to obtain the full *ftsI* gene sequence from NCBI (MG1655 genome, accession number NC_000913). This sequence was copied into ChopChop[60] to identify candidate protospacers near the N-terminus of FtsI covering residues 18 and 19 to make the sandwich fusion. The repair oligo was designed for λ-Red insertion[61] of HaloTag to break apart the protospacer, with 50 nt overlap on either end of the codons for residues 18–19 (Supplementary Table 3). Silent point mutations with comparable codon usage were also picked in this homology arm to edit the sequences of the protospacer and PAM to help select for a successful insertion of HaloTag.

**Plasmid construction**. pJM44 was constructed through using InFusion Cloning. Briefly, a pACYC-sgRNA backbone was obtained from Dr. Glenn Hauk and inverse PCR was used to create a backbone (Supplementary Table 3). A short oligo was designed to insert the 20 bp protospacer in front of the guide RNA sequence, and an InFusion reaction was run to insert the sequence based on 15 nt homology overlap with the template. The plasmid was then transformed into chemically competent Stellar competent cells (*E. coli* HST08) and outgrown in SOC for 1 h at 37 °C. Cells were plated on LB + 150 μg/ml chloramphenicol (CAM) and incubated overnight at 37 °C. Colonies were screened by colony PCR of the sgRNA and confirmed via sanger sequencing. The plasmid was purified via ThermoScientific's GeneJET Plasmid Miniprep Kit.

**JM136 construction**. CRISPR/Cas9 was performed with a method similar to previous work[62]. TB28 harboring pKD46 and pJM25 was grown at 30 °C overnight in LB + 60 μg/ml carbenicillin (CB) + 50 μg/ml kanamycin (KAN). Those cells were diluted the following day 1:100 in 50 ml LB + 60 μg/ml CB + 50 μg/ml KAN and grown for 2 h or until their OD$_{600}$ reached ~0.5. Once the culture hit the proper OD$_{600}$, 0.2% arabinose was added to the culture to induce Cas9 for 1 h. After induction, these cells were prepared to be electrocompetent such that fresh, concentrated *E. coli* could be immediately electroporated with 2 μl concentrated sgRNA plasmid pJM44 and 10 μl repair oligo. Cells were outgrown for 1.5 h at 30 °C in SOC media then plated on LB + 50 μg/ml CB + 50 μg/ml KAN + 150 μg/ml CAM to incubate for 2 days at 30 °C.

Colonies were screened by colony PCR of the chromosomal locus of *ftsI*. Overnight cultures were started of any hits at 37 °C in LB + 0.2% arabinose + 6% sucrose to kick out all plasmids. Subsequently, these overnights were serial-diluted to a factor of $10^{-6}$ and plated on LB plates that were then incubated overnight at 37 °C. Twenty random colonies were selected and screened to ensure all plasmids were excised. A final colony with all three plasmids kicked out would be checked by PCR once more and sequenced to ensure that the locus is correct.

**pRM027 construction**. The plasmid pRM027 (P$_{T5}$-lac::meos3.2-ftsI, aph) was constructed by amplifying *meos3.2*[63] and *ftsI* genes from pJB106 and pVS155-FtsI[40], respectively, and inserted to the linearized backbone from vector pCH027[40]

using the Infusion protocol (Clontech Inc.). The *cat* cassette was then replaced by an *aph* cassette amplified from pKD13.

**Preparing cells for single molecule tracking**. JM136 harboring pXY018 was grown overnight from a single colony at 37 °C in M9 + 150 μg/ml CAM. The following day, the culture was propagated 1:100 into fresh M9 + 150 μg/ml CAM and grown for 16 h at 25 °C. The following morning, 3 ml of culture was harvested at mid-log phase. Cells were concentrated to 100 μl and incubated with 10 nM of Janeliafluor 646 (JF646) for 30 min. After labeling, JM136 was washed three times with M9 medium without vitamins (M9$^-$) and concentrated to 50 μl.

EC812 harboring pRM027 was grown overnight from a single colony at 37 °C in LB + 150 μg/ml CAM and 50 μg/ml KAN with 0.2% L-arabinose. The culture was reinoculated 1:100 into fresh M9 media overnight at room temperature to log-phase for imaging. mEos3.2-FtsI supports cell growth and single molecule tracking at basal-level expression.

A 3% agar pad was prepared using a nanopillar chip with pillars of a diameter range of 1.2–1.4 μm and a length of 4.5 μm. After cooling for 30 min, cells were added to the agar pad and incubated for 2 min. The agar pad was then washed by adding 1 ml M9 and incubating the agar pad for 2 min. The media was aspirated off and the agar pad set out to dry at room temperature for ~20 min. The agar pad was sandwiched with a coverslip, sealed in a Bioptechs FSC2 chamber, and taken to the microscope for imaging.

**Microscope and imaging setup**. JM136 harboring pXY018 was imaged using two split channels in widefield on an Olympus IX-71 microscope. JF646 was imaged with 647 set to ~50 W/cm$^2$ and GFP-ZapA was imaged with 488 set to ~5 W/cm$^2$. The channels were split with an Optosplit II system containing 600 rdc, with a 700/55 emission filter for JF646 and 540/30 emission filter for GFP-ZapA. We used an Andor iXon 897 Ultra EM-CCD camera with an APON100×OTIRF objective (1.49 NA/oil) and engaged 1.6× optivar. Our camera's EM-Gain was turned on to 300 with a pre-amplifier gain setting on 3 and digitizer set to 16-bit. Baseline clamp was activated, with baseline offset set to 100.

**Phase contrast imaging**. TB28 and JM136 were grown overnight from single colonies at 37 °C in M9. The following day, the cultures were propagated 1:100 into fresh M9 and grown for 16 h at 25 °C until mid-log phase. Five hundred microliter of each culture was pelleted and concentrated in 50 μl. 0.5 μl of each strain was added to separate 3% M9 agar pads (pre-set for 30 min before use), sealed in a BiopTechs FSC2 chamber, and taken to the microscope for imaging. Cells were imaged with phase contrast on an Olympus IX-71 microscope with a 100×/1.30 NA Oil Ph3 objective and engaged 1.6× optivar. Images were recorded at a full 512 × 512 region on an Andor iXon Ultra Em-CCD camera (above) at 100 nm/pixel. Phase contrast images were recorded with 100 ms exposures and processed with Oufti to measure cell length[64].

**Single molecule tracking of Halo-FtsI::JF646**. All microscopy was recorded using Metamorph software. A region of 300 pixels wide and 512 pixels tall was used to capture both channels. Samples were put on the microscope and acclimated for 30–60 min before imaging began to minimize axial drifting. Twenty-five regions were selected with cells in microholes, and an automated journal would run to (1) autofocus on cells lying on the surface of the agar pad, then (2) move up into the sample 2 μm to find the Z-ring of cells in microholes and (3) autofocus on the GFP-ZapA ring. Once a focal plane was set, the journal would record a 400-frame movie, where each frame encapsulated 1 s (500 ms exposure, 500 ms dark time).

**Aligning channels to correct for chromatic aberration**. To correct for chromatic aberration, we imaged TetraSpeck fluorescent beads in both channels using a 50 ms exposure with the same 300 × 512 region. 647 remained set to ~50 W/cm$^2$. 488 was set to ~40 W/cm$^2$. To align the channels, we cropped the two using a custom Matlab script that then used the imregister function to align the 488 (ZapA-GFP) channel to the 647 channel (JF646). The dimensions of the crop and the transformation matrix from this alignment were used to crop all channels and align the GFP-ZapA channel to JF646.

**Analyzing single molecule trajectories**. Cropped 647 channels were first processed with ThunderSTORM[65], a plug-in for ImageJ[66]. Image filtering used a wavelet filter (B-spline) with an order of 3 and scale of 2.0. A local maximum was used to localize the molecules with 1.5*std(Wave.F1) used to identify peak intensity threshold and a connectivity of 8-neigborhood. Sub-pixel localization used a guassian PSF with a 3-pixel fitting radius and an initial sigma of 1 pixel. The post-processed data was filtered to exclude intensity values less than 300 and a sigma bandpass filter of 60–300 nm. All analysis thereafter used custom scripts in Matlab R2019a. The localizations were linked to trajectories using a nearest-neighbor algorithm modified from Sbalzarini and Koumoutsakos[67]. To link molecules which may have blinked across frames or left the focal plane, a time threshold of 15 frames was applied. The distance threshold was 300 nm/1 frame, which approximates to a diffusion coefficient of ~0.05 μm$^2$/s, or a max speed of 300 nm/s.

Only trajectories with a corresponding GFP-ZapA ring were chosen for the next step.

GFP-ZapA stacks served as both a marker to autofocus for imaging as well as for estimating the Z-ring diameter. Maximum-intensity projections were taken of movies with cells expressing GFP-ZapA and then subsequently aligned to the 647 channel as described above. Cells of interest were cropped out and a circle was then fit to the intensity profile of the GFP signal. Using the diameter, the real position of FtsI along the cell envelope can be back calculated and estimated. The trajectories were then manually segmented into mobile states only when segments were a minimum of four frames (4 s) long with processive displacements in one direction. The selected segments were fit to a straight line to minimize noise and classified as "processive" or "stationary" based on the classification description detailed in the following section. Processive segments were used to obtain the velocity, dwell time, and persistent length measurements.

**Segmentation classification**. The trajectories were first segmented manually (a segment contains at least four data points and with consistent noise). After segmenting trajectories (Supplementary Fig. 5), we classify whether the segments move processively or remain stationary. Included with our segmentation are a set of observables, $\{v, d, l, r\}$, where $v$ is the slope of a linear fit of the segment, $d$ is the total displacement, $l$ is the trajectory length, and $r$ is the standard deviation of all positions against the linear fit. Note these four parameters are not independent since $d = l \cdot v$. We subsequently combine the parameters to $\{v, l, R\}$, where $R = \frac{r}{d}$, a dimensionless quantity representing the relative level of the residual to displacement (a noise to signal parameter). Through manual inspection, we determined to classify segments as processive based on a threshold of $R \leq 0.4$.

On average, one nanopillar experiment in a day (a biological replicate) will yield 0–5 trajectories. Single cells can give up to 1–3 usable trajectories for analysis. The 77 total trajectories used in this paper come from 18 biological replicates and 49 total cells. They yielded 232 total segments, 139 of which were marked as processive by this $R$ threshold.

**Deconvolution of the FtsI fast population**. The data presented in Fig. 4f, g, h were representative of the fast, inactive population of FtsI from a two-population fit to a log-normal CDF (Supplementary Fig. 5b–d)[16].

$$f(x) = P\left(\frac{1}{x\sigma_1\sqrt{2\pi}}\right)e^{-\frac{1}{2}\left(\frac{\ln(x)-\mu_1}{\sigma_1}\right)^2} + (1-P)\left(\frac{1}{x\sigma_2\sqrt{2\pi}}\right)e^{-\frac{1}{2}\left(\frac{\ln(x)-\mu_2}{\sigma_2}\right)^2},$$

where $P$ is the percent makeup of the respective population. Briefly, we bootstrapped the FtsI data 200 times, iteratively fitting the CDF to obtain estimates for each fit parameter. A global fit of the mean bootstrapped values is presented in Supplementary Fig. 5b, and individual fits for each population are shown with respect to FtsZ (from Yang et al.[13]) in Supplementary Fig. 5c. We then used these parameters to overlay the raw FtsI histogram of 139 directional segments (Supplementary Fig. 5d). Using the slow population fit, we resampled the FtsI data 100 times by removing segments that would be selected for a bin determined by the slow population's percent makeup compared to the original histogram. The final histogram and scatterplots in Fig. 4f, g, h are representative plots from the resampling, where 74 segments remain that are likely to be fast-moving.

**Three-dimensional (3D) PALM single-molecule tracking and analysis**. Astigmatism-based 3D single-molecule tracking was performed on the same microscope. The 568 nm excitation laser power was set to 500 W/cm$^2$ with a 30 ms exposure time for 5000–10,000 frames of continuous acquisition. During the imaging, 0–1 W/cm$^2$ 405 nm activation light was increased stepwise and applied to activate mEos3.2-FtsI molecules. The UV power was tuned from sample to sample to maintain a low enough number of red-emitting molecules in each cell (<1 spot/frame/cell) for single-molecule localization and tracking.

Localization of single molecules was determined using the ThunderSTORM plugin in ImageJ[65,66]. Molecules with anomalous brightness or uncertainty (>3σ) were filtered out. Molecules were tracked across frames using custom Matlab scripts implementing the tracking algorithm described in Sbalzarini et al.[67]. To calculate the MSD, all trajectories longer than four frames were selected, and the squared displacements were calculated in 2D. Consecutive frames in the trajectories were used for displacement calculation. All three MSD curves were fitted by the anomalous diffusion function $\text{MSD}_{2D} = 4Dt^\alpha + D_0$ since these molecules diffuse on a 2D curved membrane surface. $D_0$ reflects the localization uncertainty under the imaging conditions.

**Single molecule tracking of PBP2b in B. subtilis**. B. subtilis strain bGS28, in which the native Pbp2B has been replaced with an IPTG-inducible, Halo-tagged Pbp2B, was imaged as described in Bisson-Filho et al.[12]. Briefly, cells were grown in CH media, Pbp2B was induced with 20 μM IPTG and labeled with 100 nM JF549 conjugated to HaloTag ligand, and cells were immobilized under an agarose pad for imaging. Cells were imaged in TIRF on a Nikon N-STORM microscope; time lapses were acquired with streaming 30 ms exposures for 1 min. Particle tracking was performed in TrackMate using the simple LAP tracker with the following settings: particle diameter was 300 nm, maximum linking distance was 300 nm, and no frame gaps were allowed. Tracks between 5 and 25 frames were analyzed further

using custom MATLAB code[12]. MSD vs. $t$ was calculated for each track, and the diffusion coefficient was computed by the MSD equation noted above.

**Single molecule tracking of FtsW in *Streptococcus pneumoniae***. *S. pneumoniae* strain IU15096 (D39 $\Delta cps$ $rpsL1$ $ftsW$-L$_0$-$ht$-P$_C$-$erm$) expresses a derivative of FtsW from its native chromosomal locus, with the carboxyl-terminus of FtsW fused to a 10-amino acid linker (L$_0$) connected to the HaloTag (HT) domain[41]. Frozen cultures were inoculated into 4 mL of brain heart infusion (BHI; BD Bacto, 237500) broth and serially diluted. Cultures were incubated statically without shaking at 37 °C in 5% CO$_2$ for 12–16 h. Following incubation, cultures with an optical density at 620 nm (OD$_{620}$) between 0.1 and 0.4 were washed in C+Y, pH 7.1 medium, and diluted to OD$_{620}$ = 0.01 in 5 ml of C+Y, pH 7.1 medium. Growth was monitored by measuring OD$_{620}$ every 45 min. While cells were growing, microscope slides (VWR, 16004-368) were cleaned with 70% (v/v) ethanol and a Gene Frame (Fisher, AB0576) was affixed. A 1.5% (w/v) mixture of agarose (Sigma BioReagent, A9414) in C+Y, pH 7.1 medium was melted, and agarose pads were constructed by filling the middle of the Gene Frame with the melted agarose mixture, covering with a second microscope slide, and placing at 4 °C for 45 min. After the agarose pad had solidified, and immediately prior to coverslip attachment, the top slide was removed, and $a \approx 2$ mm strip running down the center of the pad was cut out and removed with a clean razor blade. When the cell culture OD$_{620}$ reached ≈0.1, 500 μL of cells were transferred to a 1.5 ml tube and Janelia Fluor® 549 HaloTag ligand was added to a final concentration of 120 pM. Cells were vortexed, spun briefly to collect liquid, and incubated for 15 min at 37 °C in the dark with shaking at 300 rpm. Cells were centrifuged at $21,000 \times g$ for 5 min at room temperature (RT), washed twice with spinning in 0.9 ml of 37 °C-C+Y, pH 7.1 medium, and resuspended in 500 μl of 37 °C-C+Y, pH 7.1 medium. 1.2 μl of cells were pipetted onto coverslips (VWR, 48366-227) that had previously been soaked in a solution of 25% (v/v) concentrated HCl, 25% (v/v) sterile distilled H$_2$O, and 50% (v/v) ethanol for 12–24 h, then submerged 5× in sterile distilled H$_2$O, rinsed with fresh sterile distilled H$_2$O, and dried with lens paper (VWR, 52846-001). Cells were allowed to dry on the coverslips for 1–2 min, then the coverslip was attached to the Gene Frame with the cells aligned over the agarose pad. Slides were incubated in the dark at 37 °C for 30 min prior to imaging.

TIRFm was performed using the Deltavision OMX Super Resolution microscope (GE Systems), using a PCO.edge 4.2 (CMOS) camera (Kelheim Germany) and an Apo N60X/1.49 TIRF objective (Olympus). The laser line was 561 nm (emission filter 609–654) for FtsW-HT. DIC images were collected on a separate channel. Images were collected every 50 ms (20 FPS) for 100–200 s. The 561 nm (FtsW-HT) channel used a 21 ms exposure, 100% T, and a TIRF angle of 91.2°, while the DIC channel used a 3 ms exposure, 50% T, and an angle of 0° (epi). Deltavision SoftWoRx (GE Healthcare) was used to align the channels after image acquisition. Single cells were cropped from an image, and the FtsW-HT channel was processed in the FIJI plug-in ThunderSTORM[65,66]. Image filtering used a Wavelet filter (B-Spline) with an order of 3 and a scale of 2. The localization method was Local Maximum, with a peak intensity threshold of 1.5*std(Wave.F1) and 8-neighborhood connectivity. Sub-pixel localization used a PSF: Gaussian method with a fitting radius of 3 pixels, fitting method of Weighted Least Squares, and an Initial sigma of 1 pixel.

Images depicting single molecule localizations of FtsW-HT from a single cell were visually inspected to ensure each frame had a maximum of one localization. Additional localizations resulting from single pixel noise were manually removed. A maximum of five consecutive frames without a localization was allowed within a single trajectory. From individual cells, time averaged MSDs were calculated as noted above on trajectories longer than 15 frames

**Reporting summary**. Further information on research design is available in the Nature Research Reporting Summary linked to this article.

## Data availability

The authors declare that all other data supporting the findings of this study are available within the paper and its supplementary information files, or from the corresponding authors upon request. Source data are provided with this paper.

## Code availability

Code for analyzing single molecule data is available on the Xiao Lab Github[68]. Code for the computational modeling is available on the Liu lab Github[69].

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

## Acknowledgements

The authors would like to thank Dr. S. Shaw and members in the Xiao and S. Holden labs for helpful discussions and technical assistance, Dr. G. Hauk for sharing plasmids and the CRISPR-Cas9/λ-red recombineering cloning method, R. McQuillen for help cloning pRM027, Dr. E. Goley for help with growth curve measurements, and Dr. L. Lavis for sharing JF646. This work was supported in part by NIH GM007445 (to J.W.M.), NIH R35 GM131767 (to M.E.W.), equipment grant NIH 1S10OD024988-01 (to Indiana University Light Microscopy Imaging Center), NIH F31AI138430 (to M.M.L.), NSF GRFP DGE1144152 (to G.R.S.), NIH R01 GM086447 (to J.X.), GM125656 (subcontract to J.X.), NSF EAGER Award MCB-1019000 (to J.X.), a Hamilton Smith Innovative Research Award (to J.X.), Johns Hopkins University Startup fund (to J.L.) and Catalyst Awards (to J.L.).

## Author contributions

J.L. and J.X. conceived the study. J.X. created the artwork in Fig. 1. J.W.M. created the art/diagrams in Fig. 4a, performed FtsI tracking in microholes with help from Z.L. and B.S., and analyzed the resulting data (Fig. 4 and Supplementary Figs. 4, 5). X.Y., G.R.S., K.E.B., and M.M.L. measured diffusion coefficients in Fig. 5f. J.L. carried out theoretical modeling (Figs. 2, 3, 5 and Supplementary Figs 1, 2, 3). J.L., J.X., M.E.W., and E.G. supervised the work. All authors contributed to concept development, data analysis, and manuscript writing.

## Competing interests

The authors declare no competing interests.
