## [Peer Review File · Nature Communications]

REVIEWER COMMENTS

Reviewer #1 (Remarks to the Author):

In the previous submission, the authors already made a substantial contribution toward modelling the treadmilling dynamics of FtsZ and how they are related to other interacting divisome proteins. In the revised manuscript, the authors have carefully addressed the questions I and other reviewers raised. New simulations broaden the earlier findings, and textual revisions clarify the methods and interpretations. I think this manuscript will represent an important landmark for understanding how bacteria divide, and hopefully as new information becomes available through experiment, this model will be adapted. I believe the authors should make the source code available. That wasn't clear from the reporting summary.

Reviewer #3 (Remarks to the Author):

Having read the revised version, I still do not understand the mechanism studied by the authors.

Equations (2) and (3) are still confusing, because they do not describe what the authors actually do. The equations are continuous, whereas the position of the FtsZ-filament ends is discrete. (In this context: do you fix a different v_z at the beginning of each simulation run (l. 146)? That seems to be a strange way of doing things. Why should FtsZ filament A have a different treadmilling speed than FtsZ filament B? Rather, the addition and removal of subunits should be a random process with a fixed rate, no?)

The essential point of the model seems not to be explained in the section "Model description", but only afterwards: "As the FtsZ subunit at the shrinking end of the filament fell off, the next one in the row attracted and coupled to FtsI, which pulled FtsI to the right by ~ 5 nm." (ll 187) Even after this sentence, it remained unclear to me how the FtsI molecule gets attracted to the new leftmost FtsZ subunit of the FtsZ filament after the previous leftmost FtsZ subunit has fallen off. According to the authors' definition, the harmonic potentials extend each over the size of an FtsZ subunit. If an FtsZ subunit is taken away, according to Eq. 1, the position of the FtsI molecule does not change in this process. FtsI is now exposed to a flat potential. According to the definition of the model, only if FtsI happens to diffuse a sufficient distance to the right, it gets captured by the new leftmost FtsZ subunit.

But the above sentence and the trajectory that you present in Fig. 2B, inset suggest that you do differently: as the leftmost subunit is removed, FtsI jumps by ~ 5 nm to the right. What determines this distance? What is the molecular origin of this jump, that is, how do you justify this process? How do you determine the distance of this jump (in Fig. 2B inset, there is a jump of 10 nm)? Why should the interaction of FtsI with the leftmost FtsZ subunit right after detachment of the former leftmost FtsZ subunit be different (it must be, I guess, because only at this point there is the ~ 5 nm jump to the right of FtsI)?

In conclusion, the central element of your model is neither introduced in a sufficiently clear way (it does not appear in the section "Model description") nor does it appear to be physically coherent. For this reason, I cannot recommend publication of this work.

Reviewer #4 (Remarks to the Author):

The authors did an exemplary job in their rebuttal. I have no major or minor suggestions but to recommend publication.

It is especially pleasing to see how they addressed (and revised) their model of interdependence between PBPs diffusion and FtsZ treadmilling across bacterial species. This contribution cannot be overstated. The authors contribute with a unifying solution for 1) why in some cases FtsZ treadmilling limits cytokinesis rate (*B. subtilis*) and in others FtsI activity seems to be the bottleneck (*E. coli*); 2) other cases FtsZ treadmilling and FtsI directional motion seem to be decoupled (*S. pneumoniae*), and 3) it can interchange between different modalities presented before (*S. aureus*).

Reviewers' comments in black and responses in blue. New changes in the main text are in light blue.

Reviewer #1 (Remarks to the Author):

In the previous submission, the authors already made a substantial contribution toward modelling the treadmilling dynamics of FtsZ and how they are related to other interacting divisome proteins. In the revised manuscript, the authors have carefully addressed the questions I and other reviewers raised. New simulations broaden the earlier findings, and textual revisions clarify the methods and interpretations. I think this manuscript will represent an important landmark for understanding how bacteria divide, and hopefully as new information becomes available through experiment, this model will be adapted. I believe the authors should make the source code available. That wasn't clear from the reporting summary.

We thank the reviewer for his/her appreciation and encouragement of our work. The source code will be deposited to Github once the work is accepted for publication.

Reviewer #3 (Remarks to the Author):

Having read the revised version, I still do not understand the mechanism studied by the authors.

Equations (2) and (3) are still confusing, because they do not describe what the authors actually do. The equations are continuous, whereas the position of the FtsZ-filament ends is discrete. (In this context: do you fix a different v_z at the beginning of each simulation run (l. 146)? That seems to be a strange way of doing things. Why should FtsZ filament A have a different tread milling speed than FtsZ filament B? Rather, the addition and removal of subunits should be a random process with a fixed rate, no?)

The essential point of the model seems not to be explained in the section "Model description", but only afterwards: "As the FtsZ subunit at the shrinking end of the filament fell off, the next one in the row attracted and coupled to FtsI, which pulled FtsI to the right by ~ 5 nm." (ll 187) Even after this sentence, it remained unclear to me how the FtsI molecule gets attracted to the new leftmost FtsZ subunit of the FtsZ filament after the previous leftmost FtsZ subunit has fallen off. According to the authors' definition, the harmonic potentials extend each over the size of an FtsZ subunit. If an FtsZ subunit is taken away, according to Eq. 1, the position of the FtsI molecule does not change in this process. FtsI is now exposed to a flat potential. According to the definition of the model, only if FtsI happens to diffuse a sufficient distance to the right, it gets captured by the new leftmost FtsZ subunit.

But the above sentence and the trajectory that you present in Fig. 2B, inset suggest that you do differently: as the leftmost subunit is removed, FtsI jumps by ~ 5 nm to the right. What determines this distance? What is the molecular origin of this jump, that is, how do you justify this process? How do you determine the distance of this jump (in Fig. 2B inset, there is a jump of 10 nm)? Why should the interaction of FtsI with the leftmost FtsZ subunit right after detachment of the former leftmost FtsZ subunit be different (it must be, I guess, because only at this point there is the ~ 5 nm jump to the right of FtsI)?

In conclusion, the central element of your model is neither introduced in a sufficiently clear way (it does not appear in the section "Model description") nor does it appear to be physically coherent. For this reason, I cannot recommend publication of this work.

The reviewer raised two points. One focuses on how the stochasticity of FtsZ treadmilling speed is simulated in the model. The other touches upon the Brownian ratchet mechanism. We clarify these two points below and revised the manuscript accordingly:

1. In the model we fix the treadmilling speed of FtsZ (V_z) at the beginning of each simulation run. This treadmilling speed is drawn randomly from the experimentally measured FtsZ treadmilling speed distribution (Fig. 4F, gray bars). Although we agree with the reviewer that “*the addition and removal of (FtsZ) subunits should be a random process with a fixed rate*”, our current model treatment is necessary because of the following:
 - a. In our imaging experiment we do not have the resolution to distinguish the removal and addition of individual FtsZ subunits (each one is 5 nm). Rather, the treadmilling speed of individual filament is measured by its long-time trajectory over tens of seconds, *i.e.*, by taking the average of the instantaneous speeds of individual subunits. Importantly, this averaged speed per FtsZ filament does reflect the instantaneous speed fluctuations of individual FtsZ subunits and the associated intrinsic stochasticity of the system, because the FtsZ treadmilling speed histogram (Fig. 4F) demonstrates the associated statistical distribution of individual FtsZ filaments and preserves the stochastic effects intrinsic to the FtsZ treadmilling process.
 - b. If we wanted to take into account the explicit details of individual FtsZ subunits' dynamics, one possibility is to directly use this measured statistical speed distribution of FtsZ filaments to model the stochasticity of the on and off rates of FtsZ subunits. However, this approach is not straightforward. *In vivo* systems consume energy (ATP or GTP) and are not at/near equilibrium; *i.e.*, equilibrium statistical mechanics do not apply. For instance, if we took the $\sim 30\%$ of uncertainty and the ~ 27 nm/sec from the measured histogram as the uncertainty and the mean of FtsZ subunits' dissociation rate in our simulation, then simulated FtsZ treadmilling speed will largely converge to ~ 27 nm/sec for each individual trajectory over tens of seconds. This is because there are a few tens to hundreds of FtsZ subunit dissociation and association events that occur within this time window, and the so-averaged speed *is* the mean we put in. This suggests that either the uncertainty in the FtsZ subunit dissociation event is higher than 30%, or other stochastic processes may underlie the treadmilling of the individual FtsZ filament. These questions need to be further explored and is beyond the scope of the current manuscript. The current manuscript focuses on how FtsZ's treadmilling dynamics controls the directional movement of SPG synthesis enzymes, but not on the microscopic dynamics of how the on and off of individual FtsZ's subunits generate treadmilling.
2. The key of a Brownian ratchet is to rectify diffusion into a directed movement, which requires the diffusion to be in a proper range. After the underlying FtsZ subunit falls off from the shrinking end, the FtsI molecule diffuses; within the 5 nm the local potential is indeed flat in the eye of the FtsI molecule. However, there is a probability that the FtsI molecule could diffuse to the next FtsZ subunit in the row (*i.e.*, another ~ 5 nm to the right) and get stuck there. It could also diffuse away and dissociate from the FtsZ filament. This process is stochastic but not deterministic, which is why the FtsI's running length and duration are finite and have large distributions (Figs. 3 and 4). This physical picture is entirely consistent with our modeling and experimental results (Figs. 2A and 3B): if the diffusion of FtsI is too slow, it cannot keep up with the shrinking end of FtsZ; if it is too fast, it will be out of the control of the local asymmetric potential. Especially, it is

important that the model predicted the range of sPG synthase's diffusion constant to confer the observed directional movement, which was confirmed by our experiments (Fig. 5F). That being said, while the position of the shrinking end of the FtsZ filament jumps 5 nm every 0.2 second, the treading FtsI molecules do not – its trajectory is stochastic and fluctuates because of the diffusion step.

The reviewer also made us realize that the inset of the Fig. 2B may not be as revealing. We therefore revised Fig. 2B so that its inset has an even finer timescale of 5×10^{-6} second (e.g., our simulation timestep). It shows that FtsI does not just jump and follow the FtsZ end; instead, during the FtsZ filament shrinkage, the FtsI molecule still displays back-and-forth diffusion until it gets stuck in the next FtsZ subunit in the row. To further clarify this point, we edited the relevant text in the revised manuscript.

Reviewer #4 (Remarks to the Author):

The authors did an exemplary job in their rebuttal. I have no major or minor suggestions but to recommend publication.

It is especially pleasing to see how they addressed (and revised) their model of interdependence between PBPs diffusion and FtsZ treadmilling across bacterial species. This contribution cannot be overstated. The authors contribute with a unifying solution for 1) why in some cases FtsZ treadmilling limits cytokinesis rate (*B. subtilis*) and in others FtsI activity seems to be the bottleneck (*E. coli*); 2) other cases FtsZ treadmilling and FtsI directional motion seem to be decoupled (*S. pneumoniae*), and 3) it can interchange between different modalities presented before (*S. aureus*).

We thank the reviewer for the positive comments and constructive suggestions.

REVIEWERS' COMMENTS

Reviewer #3 (Remarks to the Author):

I thank the authors for their response; I now understand the model. The revised description of the model in the manuscript is clear, and I am happy to recommend this interesting work to be published in Nat Comms. One could certainly discuss, whether keeping a constant tread-milling speed is the best way to describe the FtzZ dynamics, but this can be done post publication.

There is one last small point that I would like to suggest to the authors: could you, please, state whether or not there is a potential barrier to cross for FtsI as it diffuses from the flat potential area into the area of the FtsZ filament with its energy landscape? Fig.1c suggests such a barrier, the text (for example, l.190) suggests otherwise.

Reviewer #3 (Remarks to the Author):

I thank the authors for their response; I now understand the model. The revised description of the model in the manuscript is clear, and I am happy to recommend this interesting work to be published in Nat Comms. One could certainly discuss, whether keeping a constant tread-milling speed is the best way to describe the FtsZ dynamics, but this can be done post publication.

There is one last small point that I would like to suggest to the authors: could you, please, state whether or not there is a potential barrier to cross for FtsI as it diffuses from the flat potential area into the area of the FtsZ filament with its energy landscape? Fig. 1c suggests such a barrier, the text (for example, l.190) suggests otherwise.

We thank reviewer 3 for his/her careful review. We have added a clarifying line in the legend of Figure 1 (Line 1100–Line 1101): “there is no energy barrier for FtsI to bind to FtsZ, because the binding potential is attractive”.